# COVID-19 vaccine prioritization of incarcerated people relative to other vulnerable groups: An analysis of state plans

**Rachel Strodel**[1]*, **Lauren Dayton**[2], **Henri M. Garrison-Desany**[3], **Gabriel Eber**[3], **Chris Beyrer**[1,2,3,4], **Joyell Arscott**[3], **Leonard Rubenstein**[3], **Carolyn Sufrin**[1,2]

**1** Johns Hopkins School of Medicine, Baltimore, MD, United States of America, **2** Department of Health, Behavior, and Society, Johns Hopkins School of Public Health, Baltimore, MD, United States of America, **3** Department of Epidemiology, Johns Hopkins School of Public Health, Baltimore, MD, United States of America, **4** Department of International Health, Johns Hopkins School of Public Health, Baltimore, MD, United States of America

\* rstrode3@jhmi.edu

## Abstract

### Background

Carceral facilities are epicenters of the COVID-19 pandemic, placing incarcerated people at an elevated risk of COVID-19 infection. Due to the initial limited availability of COVID-19 vaccines in the United States, all states have developed allocation plans that outline a phased distribution. This study uses document analysis to compare the relative prioritization of incarcerated people, correctional staff, and other groups at increased risk of COVID-19 infection and morbidity.

### Methods and findings

We conducted a document analysis of the vaccine dissemination plans of all 50 US states and the District of Columbia using a triple-coding method. Documents included state COVID-19 vaccination plans and supplemental materials on vaccine prioritization from state health department websites as of December 31, 2020. We found that 22% of states prioritized incarcerated people in Phase 1, 29% of states in Phase 2, and 2% in Phase 3, while 47% of states did not explicitly specify in which phase people who are incarcerated will be eligible for vaccination. Incarcerated people were consistently not prioritized in Phase 1, while other vulnerable groups who shared similar environmental risk received this early prioritization. States' plans prioritized in Phase 1: prison and jail workers (49%), law enforcement (63%), seniors (65+ years, 59%), and long-term care facility residents (100%).

### Conclusions

This study demonstrates that states' COVID-19 vaccine allocation plans do not prioritize incarcerated people and provide little to no guidance on vaccination protocols if they fall under other high-risk categories that receive earlier priority. Deprioritizing incarcerated people for vaccination misses a crucial opportunity for COVID-19 mitigation. It also raises

**Data Availability Statement:** All document files are available from the State COVID-19 Vaccination

Plans repository (URL: https://github.com/
rachelstrodel/State-COVID-19-Vaccination-Plans).

**Funding:** Carolyn Sufrin is supported by grants
from the NIH (NIDA-5K23DA045934-02) and the
Society of Family Planning Research Fund. The
funders had no role in study design, data collection
and analysis, decision to publish, or preparation of
the manuscript. Websites: https://www.societyfp.
org/, https://www.nih.gov/ This work is supported
in part by the Desmond M. Tutu Endowment in
Health and Human Rights. HGD and RJS are
supported by the Judy and Peter Bloom Kovler
Foundation. The funders had no role in study
design, data collection and analysis, decision to
publish, or preparation of the manuscript.

**Competing interests:** Carolyn Sufrin is an ex-
officio member of ACOG's Committee on Health
Care for Underserved Women, serving as ACOG's
liaison to the board of directors of the National
Commission on Correctional Health Care (NCCHC).
She also serves as an independent consultant/
expert witness for the non-profit organization
NCCHC Resources, Inc. This does not alter our
adherence to PLOS ONE policies on sharing data
and materials.

ethical and equity concerns. As states move forward with their vaccine distribution, further work must be done to prioritize ethical allocation and distribution of COVID-19 vaccines to incarcerated people.

## Introduction

The COVID-19 pandemic has had a dramatic and disproportionate effect on the 2.1 million incarcerated individuals in the United States (US) [1]. By June 6, 2020, the COVID-19 case rate of people in prisons was 5.5 times and the age-adjusted death rate was 3 times that of the general U.S. population [2]. Carceral facilities are sites of swift COVID-19 spread and high morbidity and mortality due to overcrowding of facilities, inadequate supply of masks, low access to sanitation, and limited or delayed medical care [3].

Beyond the heightened risk of acquiring COVID-19 due to congregate living conditions and rules and practices that are often inconsistent with public health requirements, people in carceral settings often are vulnerable to severe COVID-19 disease if they become infected. For example, in the US, there are approximately 81,600 people who are incarcerated over the age of 60 years, and older age has been found to increase risk of severe COVID-19 infection [4]. Certain underlying medical conditions can also put people at risk of COVID-19, and people in carceral settings experience high rates of chronic illness. One study found 60% of incarcerated people had at least one medical condition, and 32% had 2 or more [5]. Additionally, studies have found that incarcerated people in the US have higher rates of specific conditions that elevate risk of severe COVID-19 infection, including hypertension, cervical cancer, and liver disease [6,7].

As a result of these vulnerabilities, public health experts have called for prioritizing incarcerated individuals in the ethical and equitable allocation and distribution of safe and effective COVID-19 vaccines [8–10]. Epidemiological data and theoretical modeling have shown that infection control programs within carceral settings impact the spread of COVID-19 in the surrounding community [11]. Incarceration can also increase community spread of COVID-19: one study found that jail cycling accounted for 55% of the variance in COVID-19 case rates across Chicago zip codes early in the pandemic [12]. Due to the disproportionate criminalization and incarceration of those who are poor, Black, Latinx, Indigenous, and/or disabled, formerly incarcerated individuals often return to communities that bear a high burden of COVID-19. Therefore, coupling early vaccination in carceral settings with significantly reducing jail and prison populations is a key strategy to reduce both health inequities and viral spread in vulnerable communities [13].

With an initially limited supply of COVID-19 vaccine doses currently available, the CDC developed a three phased distribution strategy, with those in Phase 1 getting the vaccine first. The Advisory Committee on Immunization Practices (ACIP) of the Centers for Disease Control and Prevention (CDC) provides ongoing guidance on recommended COVID-19 vaccine prioritization [14]. Additionally, various groups—including the National Academies of Science, Engineering, and Medicine (NASEM), the World Health Organization, and Johns Hopkins Center for Health Security—have developed ethical allocation frameworks to suggest the relative prioritization of various groups for receiving doses of the COVID-19 vaccine [15–17]. The NASEM report's ethics-informed approach highlighted four risk-based criteria for prioritizing populations: risk of acquiring infection, risk of severe morbidity and mortality, risk of negative societal impact, and risk of transmitting infection to others.

Applying these criteria, there are several groups eligible for prioritization based on risk and workforce role. For example, both people in carceral facilities and long-term care facilities (LTCF) have risk of acquiring infection and high risk of severe morbidity and mortality. Carceral facilities are similar to LTCF—both of which are congregate living settings—in exposure to environmental vulnerabilities that increase the risk of SARS-nCoV-2 transmission [18]. This is demonstrated in the well-documented heightened risks of contracting infectious diseases such as tuberculosis [19], influenza [20], pneumonia [21,22], and HIV [23,24] in prisons and jails. Inability to socially distance, lack of sufficient personal protective equipment and cleaning supplies, dormitory-style living and shared bathrooms and common spaces all contribute to this vulnerability [3]. LTCFs experience similar issues with lack of space to distance; staff, contractors, and visitors moving between rooms and buildings; and shared accommodations, such as common areas or shared rooms. Therefore, these groups both experience high rates of infectious disease transmission once it is introduced into a facility, despite COVID-19-related infection control protocols being implemented in each. We would therefore anticipate that these groups would receive similar prioritization in an ethical and equitable vaccine allocation framework. NASEM recommends that people who are older and incarcerated or living in other congregate settings be included in Phase 1B. NASEM also assigns equal priority for older adults not in congregate settings and the general incarcerated population, suggesting both should be vaccinated in Phase 2.

Workforce function is another criterion that determines NAESM prioritization, and this has been applied to elevate allocation to certain workers. NAESM recommendations suggest that law enforcement be vaccinated in Phase 1A, in contrast to corrections workers and general population of incarcerated people which are recommended for vaccination distribution in Phase 2. The most recent ACIP recommendations at the time of this analysis deviated from NASEM guidance in prioritizing correctional staff and law enforcement during phase 1B while not mentioning people who are incarcerated [25]. It is important to note that people who are incarcerated have also worked throughout the pandemic, sometimes in high-risk positions. For example, as El Paso, TX faced a surge of COVID-19 deaths in November 2020, incarcerated people worked in morgues [26].

While these nationally-oriented criteria and frameworks serve as guidelines, states have discretion in developing their plans for COVID-19 vaccine prioritization and rollout. The CDC requested that states provide an interim plan outlining their vaccination plans by mid-October, 2020. States have used their discretion and created myriad prioritization schema that vary considerably in their assignment of groups of patients to prioritized phases. Nor are the states' plans static; many states have revised these plans since the emergency use authorization of two COVID-19 vaccines for emergency use in mid-December 2020, and one additional vaccine in late February 2021 [27–29]. Although there has been debate regarding whether people who are incarcerated should be included in COVID-19 vaccine trials [30], there has been little examination of how people in carceral settings are being prioritized by states now that effective COVID-19 vaccines are available. Some analyses in the grey literature have examined the prioritization of people who are incarcerated in state COVID-19 vaccine plans [9], but few have examined how their prioritization compares to other groups, such as those in long term care facilities (LTCF), people over the age of 65 years, correctional staff, and non-correctional law enforcement. Furthermore, there has been little discussion of whether states are including departments of corrections or correctional facilities in the development of vaccine rollout plans.

If people who are incarcerated are not appropriately included in vaccination plans, the consequences could be dire: people who are incarcerated would continue to be subjected to conditions that make them susceptible to severe COVID-19 morbidity and mortality. Furthermore,

this could exacerbate the undue impact of COVID-19 on Black, Latinx, and Indigenous people given that they disproportionately suffer from mass incarceration and racist policing practices [31]. Given that many of those incarcerated today will be released to the community within days, weeks, or months of admission, vaccination uptake in carceral settings has important ramifications for public health beyond prison or jail walls.

We conducted a policy analysis study to understand the inclusion of people who are incarcerated in vaccination plans by utilizing a rigorous method of document analysis to describe if the most recent publicly available state COVID-19 vaccine distribution plans explicitly include incarcerated individuals, and, if so, how they are prioritized relative to other populations. These results provide important insights into the nationwide vaccination strategy and priorities of state stakeholders during the fast-moving COVID-19 pandemic. The fact that many state allocation plans are in flux means that this timely analysis could motivate a reconsideration of vaccination priority if found to be inappropriate.

## Methods

To assess the relative prioritization of incarcerated people for COVID-19 vaccines, we conducted a document analysis of each state's interim COVID-19 vaccination plans. These plans were written in response to a request by the federal government that all jurisdictions funded by the CDC develop a vaccination distribution plan based on the "COVID-19 Vaccination Program Interim Playbook for Jurisdiction Operations" (herein referred to as "CDC Playbook") [32]. This analysis includes all 50 states and the District of Columbia.

The CDC Playbook requested information on 15 key areas of vaccine rollout. The present analysis examines state responses to the second and third sections, which ask states to outline "COVID-19 Organizational Structure and Partner Involvement" and "Phased approach to COVID-19 vaccination," respectively [32]. First drafts of the plans were due by October 16, 2020, with some states subsequently revising their plans. It is not always apparent if these plans apply only to incarcerated populations in state prisons, county jails, juvenile detention, or other carceral settings, unless it is specified by the state. For example, the Federal Bureau of Prisons was listed in the October 29, 2020 draft of the CDC Playbook as receiving a direct allocation of COVID-19 vaccines [33]. The present analysis includes publicly available plans and any revisions published by December 31, 2020. We also analyzed supplemental information on vaccine prioritization that was available from state health department websites, such as press releases and supplemental figures up to date as of December 31, 2020. Access to the state plans and any supplemental materials analyzed is provided in S1 Appendix.

We analyzed state vaccination allocation information using the READ approach of (1) readying materials, (2) extracting data from the documents, (3) analyzing that data, (4) distilling all findings [34]. We first collected state plans and supplemental information from health departments' websites. We developed codes *a priori* based on the CDC Playbook and CDC instructions for regional vaccination plans. Specific questions from the CDC Playbook used to generate codes and coding assumptions are listed in S2 Appendix.

Three research team members independently coded each plan for the phase at which, if at all, people who are incarcerated, correctional staff, law enforcement, people older than age 65, and people in long term care facilities were prioritized for vaccine allocation relative to the general population; as well as whether correctional agencies were included in vaccine allocation planning or as key partners for reaching critical populations. We resolved discrepancies in coding through discussion until we reached consensus [35].

## Results

In total, we analyzed 51 COVID-19 state and Washington D.C.'s vaccination plans, as well as supplemental materials from 31 states. For one state (Minnesota), only an executive summary of the state plan was publicly available.

Fig 1 shows the aggregate results of relative prioritization between incarcerated people and other vulnerable populations with similar risk. State-by-state prioritization is presented in Supplement 3. Overall, 92% of plans mentioned correctional facilities or departments of corrections as key partners for reaching critical populations (S1 Table). Forty-seven percent of states did not mention at which phase people who are incarcerated would be allocated the vaccine; 22% of states included them in Phase 1; 29% of states included them in Phase 2; and 2% of states included incarcerated people in phase 3. Meanwhile, state plans more frequently prioritized prison and jail workers (49%) and law enforcement officers (63%) for Phase 1 of vaccine distribution.

All state plans prioritized residents of LTCFs for phase 1 of vaccine distribution. Fifty-nine percent of states prioritized people who are 65 years or older for vaccination during Phase 1 of vaccine rollout.

## Discussion

Our results show that there is major variation by state in the prioritization of incarcerated people and other vulnerable populations for receipt of COVID-19 vaccination. We also demonstrate that correctional staff and law enforcement are often prioritized before incarcerated

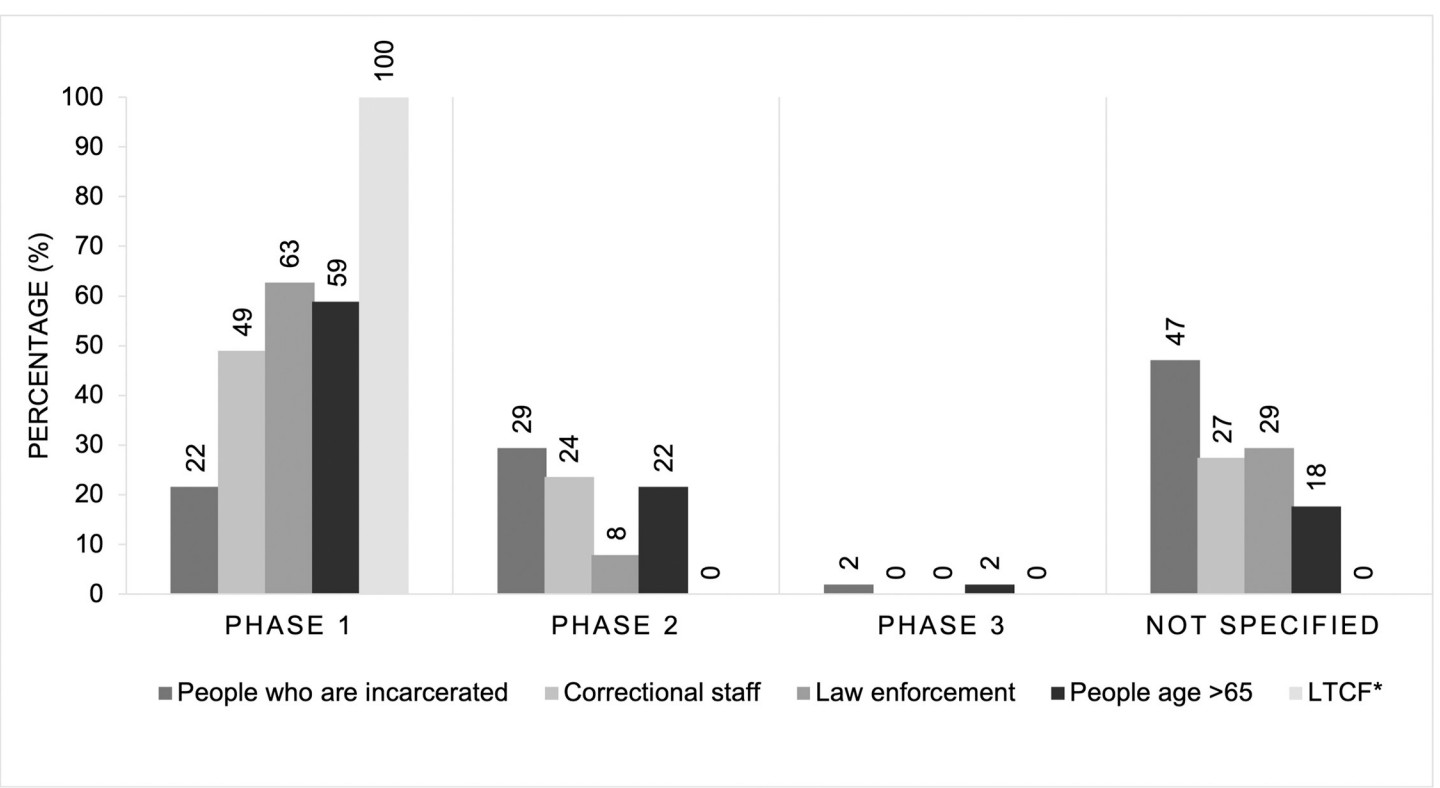

**Fig 1. Percent of incarcerated people, carceral staff, law enforcement, people >65 years, and long term care facility residents by prioritized phase in state COVID-19 plans as of December 31, 2020.** *LTCF: Long Term Care Facility.

people. While the early prioritization of correctional staff may protect people who are incarcerated by reducing community-to-facility transmission, it remains concerning that these workers are prioritized earlier for immunization despite less environmental risks incurred daily as compared to incarcerated people. Residents of LTCFs are also frequently prioritized at an earlier stage of rollout than incarcerated people, despite greater infection control challenges in carceral facilities such as a higher churn of residents entering and leaving facilities [36]. Critically, only a few states explicitly planned for vaccine distribution for older or medically vulnerable incarcerated people at the same phase as their non-incarcerated peers. This discrepancy between groups that are at increased risk of severe COVID-19 highlights a potential violation of human rights principles around the equitable access to care for incarcerated people [37].

These findings show the precarious positioning of incarcerated people in vaccination allocation strategies. Study findings identify that only about half the states specified vaccination priority for incarcerated people relative to the general population as of December 31, 2020. In those states that did, incarcerated individuals were most commonly prioritized for Phase 2 of COVID-19 vaccine rollout, consistent with the NAESM framework. An analysis conducted by the Prison Policy Initiative corroborates this finding, identifying 8 states that prioritized incarcerated people in Phase 1 of vaccine rollout, while 18 states prioritized incarcerated people in Phase 2, and 23 states did not include people who are incarcerated in their plans or had ambiguous prioritization for this group as of January 5, 2021 [38]. Our analysis expands on this work by examining not only how people who are incarcerated are prioritized, but also showing that states did not consistently apply ethical principles across groups at similar increased risk of infection and severe disease.

Our analysis also highlights a human rights violation among states which do not provide incarcerated people with the same level of care as people in the community. In accordance with The United Nations Standard Minimum Rules for the Treatment of Prisoners Rule 24: "The provision of health care for prisoners is a State responsibility. Prisoners should enjoy the same standards of health care that are available in the community, and should have access to necessary health-care services free of charge without discrimination on the grounds of their legal status." [34]. This analysis shows that incarcerated people, regardless of age, are often given lower priority than individuals 65 years and older in the community. This presents an ethical concern. Older individuals who are incarcerated represent a particularly vulnerable population, who often experience delays in routine and specialty medical care, including treatment for COVID-19, while community-dwelling older adults are able to receive care sooner, via emergency medical service providers.

The NASEM vaccine allocation framework specifies that when individuals fall into more than one phase, the earlier phase should take priority. Some states, such as North Carolina, explicitly stated that older adults or those with underlying conditions in correctional settings would receive the vaccine at an earlier stage of distribution than those who are younger and do not have underlying conditions that increase the risk for severe COVID-19. However, the vast majority of state plans lacked notable COVID-19 vaccine guidance for when individuals in overlapping categories of critical populations would receive the vaccine. This necessitates both reprioritization as well as planning to coordinate vaccine delivery in order to ensure equal distribution across groups.

For example, in several states, there are plans to partner with pharmacies, hospitals, physicians' offices, and other community-facing points of delivery to vaccinate individuals >65 years of age, who were often included in Phase 1C or a similar early phase. However, for individuals incarcerated and older than 65, there was rarely specification of whether they would be vaccinated or if they would be expected to wait until the vaccine was rolled out to all people in prison and jail facilities. This violates the ethical principle of equal concern, which guides

NASEM, WHO, and ACIP frameworks, and states that every person be concerned with similarly situated people receiving equal treatment in vaccine distribution. Therefore, in order to have synchronous roll-out for all 65+ year-olds, regardless of incarceration status, states should ensure that departments of corrections are early partners, with vaccine dissemination processes in place as soon as possible.

Our review of state partnership plans revealed that, by the end of 2020, the vast majority of states reported correctional leaders as key partners in their COVID-19 vaccination strategy. However, the terms of this partnership were often poorly defined. Some states reported that the Department of Corrections was involved in the development of the report, but often state COVID-19 vaccination plans stated that they would reach out to correctional facilities and departments as rollout progressed. Recommendations issued by experts in correctional health have called not only for administrators in correctional settings to be involved in vaccine allocation, but also people who have been or currently are incarcerated [9]. Future research should further explore the nature of public health-corrections partnerships in the rollout of COVID-19 vaccines. However, our preliminary analysis highlights a need for states to engage with a diversity of stakeholders in correctional settings early to ensure equitable vaccine allocation.

Our results also highlight the discrepancy between the policies for LTCFs and correctional facilities in vaccine dissemination planning, despite similar physical and environmental risk environments, and high rates of comorbidities. We found that people who are incarcerated were often included at a lower priority level as compared to residents of LTCFs, who have been appropriately prioritized for Phase 1 rollout in all states and Washington, D.C. Yet our results found that only 22% of states prioritized people who are incarcerated for Phase 1 of vaccine distribution, despite the high rates of chronic illnesses that place them at risk of COVID-19 infection. New Jersey defined carceral settings as a category of LTCF in their state plan, highlighting the similarities in risk that these populations face.

The early prioritization of LCTFs in the state plans reviewed for this analysis is justified: nearly 40% of COVID-19 deaths in the U.S. have been in LTCFs [39]. The failure to appropriately prioritize people who are incarcerated, however, is not. LTCFs have also received support for vaccine distribution that has not yet been afforded correctional settings. While there is a partnership between the CDC, CVS, Walgreens, and other pharmacies to ensure that LTCFs can receive on-site COVID-19 vaccinations, cold-chain support, and reporting at no additional cost, there is no indication that the same will be provided to carceral facilities [40]. Lack of support for the logistical components of vaccine rollout in carceral settings could lead to further de-prioritization of people who are incarcerated as convenience could dominate allocation decisions.

Many reports lacked clear guidance on the roll-out of vaccinations among correctional staff. During the pandemic, a number of outbreaks have been linked back to transmission among correctional staff introducing COVID-19 to the facility, which then spread among the population. Reports of COVID-19 testing among staff have shown that there are inconsistencies in where they are expected to be tested, and whether they are to be tested among community providers and report their test results to their supervisors, or whether they are to be tested through the prison medical services itself [41]. This ambiguity in protocols may also impede clarity and efficiency in vaccination rollout. Vaccine hesitancy among correctional staff may also hinder efforts to curb COVID-19 in carceral settings. In 6 states, 40% of corrections officers refused COVID-19 vaccination as of late February 2021, raising concerns that people who are incarcerated will not be protected from community-to-facility transmission unless they receive vaccination [42].

The lack of standardization of the ability of carceral facilities to provide vaccinations is another hindrance to vaccine distribution in these settings. For instance, there is wide

variability between capacity for cold chain storage and vaccine distribution between federal and state systems, individual states, prison and jail systems, and between individual facilities [3,43]. A lack of national standards for correctional medical care has necessitated a myriad of individual solutions to the dissemination problem. Future pandemic preparedness should include detailed guidance as well as steps to better equip all facilities for vaccine storage and distribution. Furthermore, at the time of analysis there was no guidance available to what extent, if any, vaccinations will be mandatory, opt-in, opt-out, or be incentivized in an ethical manner among incarcerated people, staff, and visitors. For instance, there have been reports in the past that incarcerated people received compensation for getting a flu vaccine in 2020 in Kansas, Louisiana, and Virginia [44–46]. A news report also announced the Delaware Department of Corrections will offer care packages for receiving a COVID-19 vaccine when rollout begins [47]. Any compensation to incarcerated people for vaccination should meet strict ethical standards and consideration to ensure it is not outsized or coercive, a risk that must be considered where incentives may be considerably more valued than in the community. Also, further planning will likely be necessary to create a vaccination verification system for staff and visitors to report if they have been vaccinated.

Overall, incomplete and often vague state COVID-19 vaccination plans also show the importance of clear, coherent public health communication strategies during the pandemic. A number of plans contained internal inconsistencies where groups were suggested to be prioritized for two phases at once. For instance, one state plan mentioned that "Corrections and homeless may be included in phase 1B or in phase 2" (S1 Table).

Given challenges to disseminating information to incarcerated people and visitors to the correctional facility, investment must be made in a coherent, cohesive, and robust communication strategy. The quality and mode of communication may affect vaccine hesitancy among this population and reduce uptake, ultimately leading to continued outbreaks [48]. Equity, transparency, and accountability must be critical guiding principles in communication plans. This is especially pertinent given the disregard for incarcerated people's rights and health which has, in turn, reduced trust in medical services. A recent survey of people in correctional and detention facilities found that 55% would hesitate or refuse to receive a COVID-19 vaccine, among whom 20% would not get a vaccine due to lack of trust for health care, correctional, or government personnel or institutions [49]. Thus, future work must address hesitancy barriers to vaccine uptake, including longstanding neglect and deliberate indifference to serious health needs and the ignoring of risks of harm by correctional health care systems [50–52].

## Limitations

This analysis has limitations. Primarily, the fast-moving changes in vaccine policy in light of an evolving pandemic means that it is likely states' plans will change by the time of publication. We used December 31, 2020 as the cutoff to extract updated plans for our analysis and expect that states will continue to change their recommendations as they proceed through future phases. We intend for this analysis to represent "just-in-time" public health research that can impact their future planning and goals for the COVID-19 vaccine. Reports in April 2021 that only 20% of people in federal and state prisons had received a COVID-19 vaccine, as compared to 40% of the general U.S. population, suggest these findings remain relevant to COVID-19 allocation policy [53].

Additionally, not all COVID-19 vaccination plans were publicly available—for one state, only an executive summary could be used. This analysis also does not clarify how different incarcerated populations—such as those in immigration detention centers, jails, and state and

federal prisons—may differ in priority. Therefore, these results give an overall view of state-wide vaccine policies but may not be representative of the nationwide incarcerated population. Finally, the omission of people who are incarcerated from state plans may be attributable to a lack of specificity rather than a deliberate de-prioritization of these individuals. Yet given the complex challenges of distributing even routine vaccines in carceral settings, these omissions may still result in vaccination delays for incarcerated persons.

### Strengths

Our study made use of a triple-coding strategy and iterative process for updating plans and related materials through December 31, 2020. It was important to use such a detailed coding strategy given the variation in formats and detail included across states, and the real impact of these changing plans as vaccines became available at the end of December 2020. This allowed for accurate coding of the most recent statewide plans and adds to the rigor of our findings. Due to the CDC's requirement that each state upload a COVID-19 vaccination plan, we had data available for all 50 states and the Washington D.C, providing a nationally representative document review to compare and contrast vaccination plans.

### Conclusions

State plans for COVID-19 vaccine rollout do not prioritize people who are incarcerated to the same extent as other groups with similar levels of risk for acquiring COVID-19 and experiencing morbidity and mortality due to this virus. This dissonance presents an opportunity: since current vaccination plans are in flux, states have a window to re-assign people who are incarcerated to levels of priority commensurate with their risk. Doing so will require transparent communication with affected individuals, as well as increased coordination with correctional facilities, departments of corrections, and other stakeholders. States that fail to prioritize people who are incarcerated deny them access to fundamental health care and place them at risk of morbidity, long-term sequelae, and mortality.

### Supporting information

**S1 Table. Coding of state's relative prioritization of people who are incarcerated, correctional staff, law enforcement, people ≥65, and LTCF residents.**
(DOCX)

**S1 Appendix. List of state plans and supplementary documents analyzed.**
(DOCX)

**S2 Appendix. Code development and coding assumptions.**
(DOCX)

### Author Contributions

**Conceptualization:** Rachel Strodel, Lauren Dayton, Gabriel Eber, Chris Beyrer.

**Data curation:** Rachel Strodel, Lauren Dayton, Henri M. Garrison-Desany.

**Formal analysis:** Rachel Strodel, Lauren Dayton, Henri M. Garrison-Desany, Gabriel Eber.

**Methodology:** Rachel Strodel, Lauren Dayton, Henri M. Garrison-Desany, Carolyn Sufrin.

**Visualization:** Lauren Dayton.

**Writing – original draft:** Rachel Strodel, Lauren Dayton, Henri M. Garrison-Desany, Gabriel Eber, Carolyn Sufrin.

**Writing – review & editing:** Rachel Strodel, Lauren Dayton, Henri M. Garrison-Desany, Gabriel Eber, Chris Beyrer, Joyell Arscott, Leonard Rubenstein, Carolyn Sufrin.

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
