## [Decision Letter · Decision Letter 0]

7 Apr 2021

PONE-D-21-04563

COVID-19 vaccine prioritization of incarcerated people relative to other vulnerable groups: An analysis of state plans

PLOS ONE

Dear Dr. Strodel,

Thank you for submitting your manuscript to PLOS ONE. After careful consideration, we feel that it has merit but does not fully meet PLOS ONE’s publication criteria as it currently stands. Therefore, we invite you to submit a revised version of the manuscript that addresses the points raised during the review process.

We greatly appreciate your patience with peer reviews, and are confident that the manuscript will benefit from the minor revisions recommended below.

We look forward to receiving your revised manuscript.

Kind regards,

Andrea Knittel

Academic Editor

PLOS ONE

Journal Requirements:

'I have read the journal's policy and the authors of this manuscript have the following competing interests: Carolyn Sufrin is an ex-officio member of ACOG’s Committee on Health Care for Underserved Women, serving as ACOG’s liaison to the board of directors of the National Commission on Correctional Health Care (NCCHC). She also serves as an independent consultant/expert witness for the non-profit organization NCCHC Resources, Inc. '

a. Please confirm that this does not alter your adherence to all PLOS ONE policies on sharing data and materials, by including the following statement: "This does not alter our adherence to  PLOS ONE policies on sharing data and materials.” (as detailed online in our guide for authors http://journals.plos.org/plosone/s/competing-interests).  If there are restrictions on sharing of data and/or materials, please state these.

Please note that we cannot proceed with consideration of your article until this information has been declared.

Reviewers' comments:

Reviewer's Responses to Questions

**Comments to the Author**

1. Is the manuscript technically sound, and do the data support the conclusions?

Reviewer #1: Yes

Reviewer #2: Yes

2. Has the statistical analysis been performed appropriately and rigorously? 

Reviewer #1: N/A

Reviewer #2: Yes

3. Have the authors made all data underlying the findings in their manuscript fully available?

Reviewer #1: Yes

Reviewer #2: No

4. Is the manuscript presented in an intelligible fashion and written in standard English?

Reviewer #1: Yes

Reviewer #2: Yes

5. Review Comments to the Author

Reviewer #1: This is a well written and important article on covid vaccine distribution policy in the U.S.

A few minor comments:

Line 198: "We also demonstrate that correctional staff are often prioritized before incarcerated people, despite similar environmental risks incurred daily." You could state here the importance of C.O.s due to their contact with the community and potential to introduce new infections from the community to the correctional setting and vice versa. However, I would potentially rephrase to "... despite less environmental risk..." because C.O. are not constantly in the same congregate living conditions and have better access to hygiene materials.

Line 249: The U.N. rule is repeated and sentence could be more concise to avoid redundancy.

Reviewer #2: This is an important contribution to the literature on COVID-19 in carceral settings and the lack of prioritization of this vulnerable population.

Abstract:

- Do you have space in the abstract to also explicitly say the number of states that have placed incarcerated populations in group 3 priority?

Introduction:

- Limited vaccine availability is mentioned as a motivation. Given that the number of vaccines available has greatly increased in the past few months, I’m wondering if you could include something about how this is even a more pressing now that vaccines are not in as short of supply and incarcerated populations remain under-prioritized.

- The interpretation of case rates should be 5.5 times that of vs. 5.5 times higher (line 56).

- When discussing community spread in the introduction, could you add that communities that incarcerated individuals go home to bear a disproportionate burden of COVID-19? This brings together the need to both decarcerate and prioritize vaccination – they cannot happen separately.

- You discuss workforce exposure but neglect to mention that incarcerated persons have worked throughout the pandemic – sometimes in jobs that in direct contact with COVID-19 (e.g., burial services). It would be helpful to include this piece as well.

Methods:

- While listed in the limitations section, I would add in the methods section that it is unclear if this analysis captures all incarcerated persons or only a subset. Given that it is state wide guidance, I would think that this most likely captures those in state prisons rather than county jails, federal prisons, juvenile detention, etc.

Results:

- While the results are informative, I would be interested to see a sort of cross-tab. For instance, in states where prison and jail employees are prioritized, are those states also prioritizing incarcerated individuals? How much do these metrics track together – as this has clear implications for COVID-19 spread within these facilities.

- Similarly, it seems that, substantively, prioritizing 65+ and long-term facility care residents is slightly different than states prioritizing prison and jail employees and law enforcement, as the latter group’s prioritization directly affects COVID-19 spread in carceral settings. I would recommend framing the findings in this way.

- Please list data sources in appendix (e.g., website links).

Discussion:

- There is no mention of vaccine hesitancy among staff and incarcerated individuals, which has recently come to the forefront of many discussions (e.g., incarcerated persons with allergies may not trust the medical providers administering the vaccines to quickly respond to adverse effects; many correctional staff say that they will not take the vaccine; incarcerated persons have historically been used to test medicines and vaccines and rightly mistrust medical and public health systems). This seems important to discuss.

6. PLOS authors have the option to publish the peer review history of their article (what does this mean?). If published, this will include your full peer review and any attached files.

Reviewer #1: No

Reviewer #2: **Yes: **Katherine LeMasters

---

## [Author Response · Author response to Decision Letter 0]

10 May 2021

Line 198: "We also demonstrate that correctional staff are often prioritized before incarcerated people, despite similar environmental risks incurred daily." You could state here the importance of C.O.s due to their contact with the community and potential to introduce new infections from the community to the correctional setting and vice versa. However, I would potentially rephrase to "... despite less environmental risk..." because C.O. are not constantly in the same congregate living conditions and have better access to hygiene materials.

Thank you for this edit; we have revised the manuscript accordingly to state: “While the early prioritization of correctional staff may protect people who are incarcerated by reducing community-to-facility transmission, it remains concerning that these workers are prioritized earlier for immunization despite less environmental risks incurred daily as compared to incarcerated people.”

Line 249: The U.N. rule is repeated and sentence could be more concise to avoid redundancy.

We appreciate this recommendation and have removed one of the sentences on the UN rule to reduce redundancy. 

Reviewer #2: This is an important contribution to the literature on COVID-19 in carceral settings and the lack of prioritization of this vulnerable population.

Abstract:

- Do you have space in the abstract to also explicitly say the number of states that have placed incarcerated populations in group 3 priority?

Yes. Thank you for this recommendation. We have added this statistic to the abstract: “We found that 22% of states prioritized incarcerated people in Phase 1, 29% of states in Phase 2, and 2% in Phase 3, while 47% of states did not explicitly specify in which phase people who are incarcerated will be eligible for vaccination.”

Introduction:

- Limited vaccine availability is mentioned as a motivation. Given that the number of vaccines available has greatly increased in the past few months, I’m wondering if you could include something about how this is even a more pressing now that vaccines are not in as short of supply and incarcerated populations remain under-prioritized.

Thank you for this comment. We now state “Reports in April 2021 that only 20% of people in federal and state prisons had received a COVID-19 vaccine, as compared to 40% of the general U.S. population, suggest these findings remain relevant to COVID-19 allocation policy.[49]”

- The interpretation of case rates should be 5.5 times that of vs. 5.5 times higher (line 56).

Thank you for this revision. We have edited the case rate accordingly to now read: “By June 6, 2020, the COVID-19 case rate of people in prisons was 5.5 times and the age-adjusted death rate was 3 times that of the general U.S. population.[2]”

- When discussing community spread in the introduction, could you add that communities that incarcerated individuals go home to bear a disproportionate burden of COVID-19? This brings together the need to both decarcerate and prioritize vaccination – they cannot happen separately.

Thank you for this revision. We have revised this accordingly and added a citation to support this. “Due to the disproportionate criminalization and incarceration of those who are poor, Black, Latinx, Indigenous, and/or disabled, formerly individuals often return to communities that bear a higher burden of COVID-19. Therefore, coupling early vaccination in carceral settings with significantly reducing jail and prison populations are key strategies to reduce both health inequities and viral spread in vulnerable communities.[12]”

- You discuss workforce exposure but neglect to mention that incarcerated persons have worked throughout the pandemic – sometimes in jobs that in direct contact with COVID-19 (e.g., burial services). It would be helpful to include this piece as well.

Thank you for this revision. We have added this to the manuscript. “It is important to note that people who are incarcerated have also worked throughout the pandemic, sometimes in high-risk positions. For example, as El Paso, TX faced a surge of COVID-19 deaths in November 2020, incarcerated people worked in morgues.[25]”

Methods:

- While listed in the limitations section, I would add in the methods section that it is unclear if this analysis captures all incarcerated persons or only a subset. Given that it is state wide guidance, I would think that this most likely captures those in state prisons rather than county jails, federal prisons, juvenile detention, etc.

Thank you for this revision. We have added this revision and further clarification on the fact that the analysis does not apply to Federal BOP populations: “It is not always apparent if these plans apply only to incarcerated populations in state prisons, county jails, juvenile detention, or other carceral settings, unless it is specified by the state. For example, the Federal Bureau of Prisons was listed in the October 29, 2020 draft of the CDC Playbook as receiving a direct allocation of COVID-19 vaccines.[32]”

Results:

- While the results are informative, I would be interested to see a sort of cross-tab. For instance, in states where prison and jail employees are prioritized, are those states also prioritizing incarcerated individuals? How much do these metrics track together – as this has clear implications for COVID-19 spread within these facilities.

We agree that a cross-tab would be informative in understanding whether nor not correctional staff are prioritized at a similar phase as incarcerated people. However, we believe this analysis is outside the scope of this study. This analysis is primarily a qualitative document analysis and we believe that it would not be methodologically appropriate to perform quantitative analyses of this nature on our coding results. We believe figure 1 clearly demonstrates the disparity in prioritization between people who are incarcerated and carceral staff. 

We have added the following revision to the discussion to further explore the implications of how correctional staff and incarcerated people are being prioritized: “While the early prioritization of correctional staff may protect people who are incarcerated by reducing community-to-facility transmission, it remains concerning that these workers are prioritized earlier for immunization despite less environmental risks incurred daily as compared to incarcerated people.”

- Similarly, it seems that, substantively, prioritizing 65+ and long-term facility care residents is slightly different than states prioritizing prison and jail employees and law enforcement, as the latter group’s prioritization directly affects COVID-19 spread in carceral settings. I would recommend framing the findings in this way.

We have edited the framing of this finding in the first paragraph of the discussion to highlight how prioritizing correctional staff/law enforcement could impact spread in carceral settings. (See the quotation mentioned in Reviewer 1’s comment regarding Line 198.)

- Please list data sources in appendix (e.g., website links).

Thank you for this recommendation. Due to the fact that URLs for state vaccine plans are continually being updated and do not necessarily represent the actual data analyzed, we have created a publicly-accessible folder with all the plans and supplements analyzed. A link to this folder is now included in Supplement 1.

Discussion:

- There is no mention of vaccine hesitancy among staff and incarcerated individuals, which has recently come to the forefront of many discussions (e.g., incarcerated persons with allergies may not trust the medical providers administering the vaccines to quickly respond to adverse effects; many correctional staff say that they will not take the vaccine; incarcerated persons have historically been used to test medicines and vaccines and rightly mistrust medical and public health systems). This seems important to discuss.

Thank you for this recommendation. Vaccine hesitancy or outright refusal to accept vaccination are important issues that deserve further study. Regrettably this document review, does not have data regarding these issues among correctional staff or incarcerated persons but we believe that additional quantitative and qualitative research on vaccine hesitancy or refusal are valuable areas of future study which we call for in this manuscript. 

We mention vaccine hesitancy briefly in the initial manuscript (last paragraph of discussion) but have now added additional commentary to highlight the importance of this topic, including the following:

“Vaccine hesitancy among correctional staff may also hinder efforts to curb COVID-19 in carceral settings. In 6 states, 40% of corrections officers refused COVID-19 vaccination as of late February 2021, raising concerns that people who are incarcerated will not be protected from community-to-facility transmission unless they receive vaccination.[41]”

And:

“Equity, transparency, and accountability must be critical guiding principles in communication plans. This is especially pertinent given the disregard for incarcerated people's rights and health which has, in turn, reduced trust in medical services. A recent survey of people in correctional and detention facilities found that 55% would hesitate or refuse to receive a COVID-19 vaccine, among whom 20% would not get a vaccine due to lack of trust for health care, correctional, or government personnel or institutions.[48] Thus, future work must address hesitancy barriers to vaccine uptake, including longstanding neglect and deliberate indifference to serious health needs and the ignoring of risks of harm by correctional health care systems.[49–51]"

---

## [Decision Letter · Decision Letter 1]

31 May 2021

COVID-19 vaccine prioritization of incarcerated people relative to other vulnerable groups: An analysis of state plans

PONE-D-21-04563R1

Dear Dr. Strodel,

We’re pleased to inform you that your manuscript has been judged scientifically suitable for publication and will be formally accepted for publication once it meets all outstanding technical requirements.

Kind regards,

Andrea Knittel

Academic Editor

PLOS ONE
---

## [Editor Report · Acceptance letter]

7 Jun 2021

PONE-D-21-04563R1 

COVID-19 vaccine prioritization of incarcerated people relative to other vulnerable groups: An analysis of state plans 

Dear Dr. Strodel:

I'm pleased to inform you that your manuscript has been deemed suitable for publication in PLOS ONE. Congratulations! Your manuscript is now with our production department. 

Kind regards, 

on behalf of

Dr. Sungwoo Lim 

Academic Editor

PLOS ONE